# Syndesmotic fixation in Weber B ankle fractures: A systematic review

**Brandon Lim**[1]*, **Mohamed Shaalan**[2], **Sven O'hEireamhoin**[2], **Frank Lyons**[2]

**1** School of Medicine, Trinity College Dublin, Dublin, Ireland, **2** Department of Trauma and Orthopaedic Surgery, Mater Misericordiae University Hospital, Dublin, Ireland

* limb@tcd.ie

**Data Availability Statement:** All relevant data are within the manuscript and its Supporting Information files.

## Abstract

Weber Type B fractures often arise from external rotation with the foot supinated or pronated. Altered tibiofibular joint kinematics in Weber B fractures are responsible for syndesmotic damage seen in Weber B fractures. Weber B fractures are managed using open reduction and internal fixation if displaced. The syndesmosis is injured in up to 40% of cases resulting in an unstable injury with a syndesmotic diastasis. This systematic review aimed to evaluate the current literature on syndesmotic fixation in Weber B fractures, assess the outcomes and complications of syndesmotic fixation and assess the necessity of syndesmotic fixation in Weber B fractures. A search was carried out on the EMBASE, PubMed and CINAHL databases and eight studies assessing the outcomes of syndesmotic fixations versus no syndesmotic fixation with 292 Weber B ankle fractures were included in this systematic review. Results showed significant heterogeneity so a narrative review was conducted. Results of these studies showed that functional, radiological, and quality-of-life outcomes and incidences of post-traumatic osteoarthritis in patients with syndesmotic screws were similar to those of patients not managed with syndesmotic screws. Only one favoured syndesmotic fixation in all cases of diastasis. As such, syndesmotic fixation with screws may not be necessary in the management of Weber B fractures. Screws are also associated with breakage, loosening, local irritation and infections. Suture button devices and antiglide fixation techniques appear to be valid alternatives to syndesmotic screws. It was found that there was no need for routine hardware removal unless the hardware was causing significant side effects for the patient.

## Introduction

The Weber classification of ankle fractures is the most used classification system and is based on the level of the fibular fracture relative to the tibiofibular syndesmosis [1]. Weber B fractures occur at the level of the syndesmosis and where a long oblique fracture may also extend above the syndesmosis [2]. Ankle fractures most frequently occur in young men and elderly women while fracture-dislocations usually occur in severe trauma [3]. Weber Type B fractures often arise from external rotation with the foot supinated (SER) or pronated (PER), hence they also correspond with Lauge-Hansen SER IV ankle fractures [4]. Weber B fractures are the

**Funding:** The author(s) received no specific funding for this work.

**Competing interests:** The authors have declared that no competing interests exist.

most common injury treated in orthopaedic surgery and are managed using open reduction and internal fixation (ORIF) if displaced [5, 6]. Altered tibiofibular joint kinematics in Weber B fractures are responsible for syndesmotic damage seen in Weber B fractures [3]. The syndesmosis is injured in up to 40% of cases resulting in an unstable injury with a syndesmotic diastasis [4]. It is agreed upon that ankle stability relies on maintaining the integrity of the lateral column of the ankle [7]. Weber B fractures are associated with post-traumatic osteoarthritis and limb-threatening complications [3].

Syndesmotic fixation (SF) using screws has been most successful in the management of syndesmotic injuries but is prone to breaking or loosening, requiring routine hardware removal that increases the risk of infection and a loss of syndesmotic reduction [3]. Although implantable suture buttons allow micromotion and potentially eliminate the need for hardware removal while matching the mechanical strength of screws, they may also result in hardware irritation, infection, osteomyelitis, and osteolysis [3]. Furthermore, a study of 238 Weber B fractures post-lateral malleolar fixation identified 93 cases of syndesmotic instability [8]. The CROSSBAT trial also found that Weber B fractures treated surgically were associated with more adverse events than those managed non-surgically [9].

However, some studies have found no difference in clinical outcomes of unstable Weber B fractures that were treated operatively versus non-operatively [10, 11]. There is thus still uncertainty regarding the necessity of operative management of Weber B fractures and a gap in the literature regarding the need for syndesmotic fixation in the management of Weber B fractures. As such, there is a need to evaluate whether fixation of the syndesmosis in the management of Weber B fractures has any meaningful impact on patient functional, radiological, and quality-of-life outcomes as well as incidences of post-operative complications. This systematic review aims to: 1) evaluate the current literature on syndesmotic fixation in Weber B fractures; 2) evaluate the outcomes of SF vs non-SF management in Weber B fractures; 3) evaluate the complications of SF vs non-SF management in Weber B fractures; and 4) assess the necessity of syndesmotic fixation in Weber B fractures.

## Methods

The protocols for this systemic review are in adherence to the guidelines prescribed by the Preferred Reporting Items for Systematic Reviews and Meta-Analyses (PRISMA) [12].

### Search strategy

We searched EMBASE, PubMed and CINAHL from their respective inceptions through 6 November 2023 for studies that examined Weber B ankle fractures and syndesmotic repair. Databases were searched using the search terms under two themes and combined using the Boolean operator 'AND'. Terms used for the theme 'Weber B ankle fractures' were: "ankle fracture*" OR "Weber type B fracture*" OR and "Weber B fracture*". Terms used for the theme 'syndesmotic fixation' were: "syndesmotic fixation" OR "syndesmotic repair" OR "syndesmotic screw". Search results were imported into the Covidence online software tool and duplicates were automatically removed.

### Eligibility criteria

During title and abstract screening, animal studies, case reports, technical reports, thesis papers, systematic reviews, letters to the editor, and conference abstracts were excluded. Studies that did not address the outcome of syndesmotic repair of Weber B fractures were excluded. During full-text screening, only full-length manuscripts focussed on the outcome of syndesmotic repair of Weber B fractures were included.

## Study selection

Studies were uploaded as abstracts to the online systematic review management platform Covidence (Melbourne, Australia) from which duplicate studies were automatically identified and deleted. Two reviewers evaluated the retrieved studies independently. Discrepancies were resolved by discussion between the two reviewers.

## Data abstraction

Studies identified as relevant after full-text review underwent data abstraction. Two reviewers independently extracted and recorded data using offline Microsoft Excel sheets. Relevant data included study design, sample sizes, method of syndesmotic repair, follow-up duration, clinical and radiological outcomes, types of complications, and a rating of each study's methodological quality.

## Assessment of the quality of studies

Two reviewers used the Newcastle-Ottawa Scale (NOS) to assess the methodological quality and risk of bias in eligible studies [13]. This system assesses the quality of cohort and case-control studies based on selection, comparability, and outcome/exposure criteria. Scores for NOS assessing cohort studies range from zero to nine. Studies with higher NOS scores indicate a lower risk of bias.

The following variables were identified to likely confound associations between repair of the syndesmosis and outcome measures: 1) body mass index; 2) additional fractures or ligamentous injuries; and 3) previous operations to the injured ankle. Observational studies that attempted to control for one or more of these theoretical confounds were at a decreased risk of comparability bias based on NOS criteria.

## Statistical analysis

Studies included in this systematic review displayed significant heterogeneity, in terms of syndesmotic fixation techniques, follow-up durations, and primary and secondary outcome measures. As such, a meta-analysis of the data from these studies was deemed inappropriate. Therefore, a narrative review is presented instead.

# Results

The literature search from all databases provided 340 potentially relevant articles from EMBASE (n = 155), PubMed (n = 131), and CINAHL (n = 54) (Fig 1). After excluding 156 duplicates, 184 records were further analysed. 80 publications were removed after reviewing titles and abstracts. Of the residual corpus of literature, a full-text review was performed which identified eight relevant studies for inclusion.

## Study characteristics

The characteristics of the studies selected for qualitative synthesis can be found in Table 1. Sample sizes ranged from 6 [14] to 68 [15]. Of the eight studies reviewed, five were prospective cohort studies [5, 14, 16–18], and three were retrospective cohort studies [6, 15, 19]. Out of eight studies, seven used a mix of patient interviews, questionnaires, and clinical assessments [5, 6, 14, 15, 17–19], six utilised plain radiography [5, 6, 14–17], and three utilised magnetic resonance imaging (MRI) [5, 6, 19]. The Olerud-Molander-Ankle-Score (OMAS) was used in four studies [5, 6, 17, 18], the RAND 36-Item Health Survey to assess quality of life in three studies [5, 17, 18], the 100-mm visual analogue scale (VAS) for ankle function and pain in

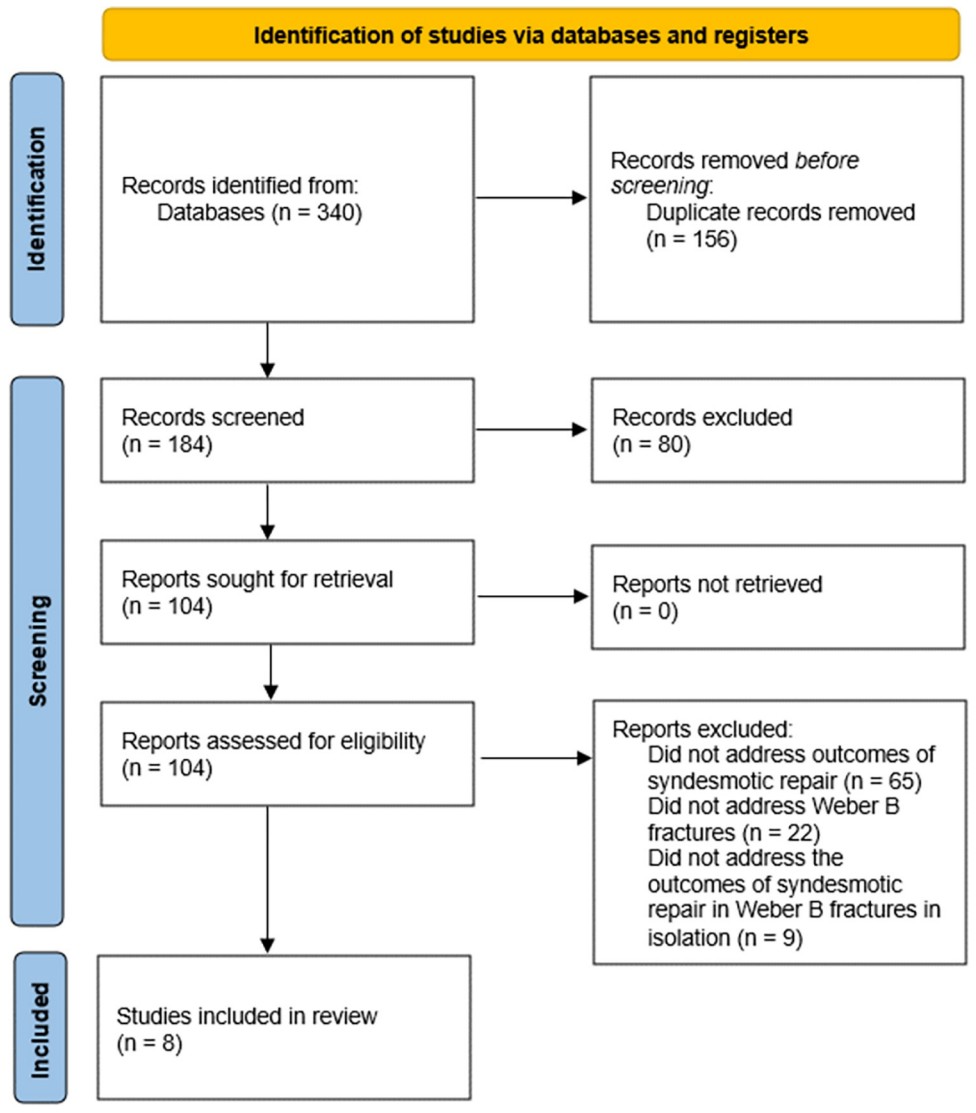

**Fig 1. PRISMA flowchart.**

three studies [5, 17, 18], the Hannover Score in two studies [6, 14], the American Orthopedic Foot and Ankle Society (AOFAS) scoring system in two studies [6, 19], the Short Form Health Survey (SF)-36 v2.0 questionnaires to evaluate health-related quality of life in two studies which included the Physical and Mental Component Summary (PCS, MCS) [6, 19], the Foot-Function-Index (FFI) in one study [6], and the Foot and Ankle Ability Measure (FAAM) questionnaire Activities of Daily Living (ADL) and Sports subscales in one study [15]. Three studies specified utilising a goniometer to assess the range of motion (ROM) [5, 6, 18]. Post-operative arthritis was assessed using the Kellgren-Lawrence Score (KLS) in three studies [5, 6, 18].

## Assessment of syndesmotic instability

Kohake et al. evaluated the stability of the syndesmosis using an intraoperative hook test to pull the fibula laterally, defining instability as a widening of >2mm [6]. Van den Bekerom et al. also utilised the intraoperative bone hook test, under fluoroscopic guidance if necessary,

**Table 1. Characteristics of studies.**

| Author (year) | Origin | Study Design | Sample size (n) | SF (n) | No SF (n) | Follow-up | Primary Outcomes | Secondary Outcomes |
|---|---|---|---|---|---|---|---|---|
| Kohake et al., (2019) [6] | Germany | Retrospective cohort study. | 61 | 21 | 40 | Mean 6.6 years (range 2–12). | AOFAS, FFI, HS, OMAS. | SF-36 PCS MCS, VAS (pain) post-operative osteoarthritis assessed using the KLS. |
| Kortekangas et al., (2014) [5] | Finland | Prospective cohort study. | 24 | 13 | 11 | Mean 58 months. | OMAS, RAND 36, VAS (function and pain), ROM, tibiotalar clear space, tibiofibular clear space, congruence of the tibiotalar joint. | - |
| Lehtola et al., (2022) [18] | Finland | Prospective cohort study. | 24 | 13 | 11 | Mean 9.7 years. | OMAS. | RAND 36, VAS (function and pain), ROM, radiography (talocrural joint osteoarthritis, ankle mortise congruity). |
| Ntalos et al., (2018) [19] | Germany | Retrospective cohort study. | 49 | 30 | 19 | Mean 34.5 months. | AOFAS, SF-36, tibiofibular alignment. | - |
| Paez et al., (2021) [15] | USA | Retrospective cohort study. | 68 | 19 | 49 | Mean 4.35 years. | FAAM questionnaire, SANE questionnaire. | Return to sport, patient satisfaction scores, complication rates. |
| Pakarinen et al., (2011) [17] | Finland | Prospective cohort study. | 24 | 13 | 11 | 1 year. | OMAS, RAND 36, VAS (function and pain), ROM, tibiotalar clear space, tibiofibular clear space, congruence of the tibiotalar joint. | - |
| Sipahioglu et al., (2018) [14] | Turkey | Prospective cohort study. | 6 | 6 | 0 | Mean 49 months. | HS, anteroposterior tibiofibular distance, lateral fibular distance, and medial mortise distance. | - |
| van den Bekerom et al., (2013) [16] | Netherlands | Prospective cohort study. | 36 | 36 | 0 | Mean 3.4 years. | Radiographic complications. | - |

Table 1 Abbreviations: AOFAS = Foot and Ankle Society scoring system; AP = anteroposterior; FAAM ADL = Foot and Ankle Ability Measure Activities of Daily Living Subscale (possible score 0% to 100%); FAAM Sports = Foot and Ankle Ability Measure Sports Subscale (possible score 0% to 100%); FFI = Foot-Function-Index; HS = Hannover-Score; KLS = Kellgren-Lawrence Score; OMAS = Olerud-Molander-Ankle-Score; RAND 36 = RAND 36-Item Health Survey (physical and pain); ROM = range of motion; SANE = Single Assessment Numerical Evaluation; SF = syndesmotic fixation; SF-36 PCS MCS = Short Form Health Survey (SF)-36 v2.0 Physical and Mental Component Summary; VAS = 100-mm visual analogue scale for ankle function and pain.

and an image intensifier to assess syndesmotic stability [16]. Paez et al. assessed instability by applying force to the distal fibula, defining instability as a >2mm of lateral translation of the fibula with widening of the ankle mortise on the anteroposterior view or >2mm of posterior translation of the fibula relative to the tibia on a lateral view [15]. Kortekangas et al., Lehtola et al., and Pakarinen et al. utilised external rotation stress tests, defining a positive result as a difference of >2mm side-to-side in the tibiotalar or tibiofibular clear spaces on mortise radiographs [5, 17, 18]. Ntalos et al. evaluated syndesmotic instability with an external rotation stress test, the bone hook test or direct visualisation of the syndesmosis [19]. Sipahioglu et al. evaluated tibiofibular diastasis using three-plane ankle radiographs (anteroposterior, lateral, and mortise) taken bilaterally for comparison [14].

## Surgical management

All eight studies utilised syndesmotic screws [5, 6, 14–19]. Paez et al. also assessed the use of suture-button constructs although the exact number of patients with Weber B fractures that were managed with a suture-button construct is unclear [15]. Kohake et al. managed unstable distal fibulas with a screw placed parallel to the syndesmosis [6]. Lehtola et al. and Pakarinen et al. managed lateral malleolus fractures with two 3.5-mm cortical screws or a one-third

tubular plate with or without a lag screw, medial malleolus fractures with two partially-threaded 3.5mm cancellous screws, and posterior malleolus fracture involving >30% of the articular surface with 3.5mm partially-threaded cancellous screws from anterior to posterior [17, 18]. Ntalos et al. managed Weber B fractures using a one-third tubular plate, medial malleolus fractures with two screws or a combination of screw and Kirschner-wire, posterior malleolus fractures with two screws fixed from the front, and fracture-dislocations with closed reduction with or without external fixation followed by ORIF [19]. Paez et al. managed fibular fractures with one or two lag screws and a neutralization plate, non-comminuted long oblique fractures with two to three lag screws and no neutralization plate, medial malleolus fractures with one or two 4-mm screws, and posterior malleolus fractures involving >20% of the posterior articular surface with 4-mm screws [15]. Sipahioglu et al. managed medial malleolus fractures using a 4.0mm malleolar screw with or without a washer, fibular fractures with plate fixation, tibiofibular diastasis using 4.0mm malleolar screws placed parallel and 2-3cm above the ankle joint and 20–30˚ anteromedially, and syndesmosis fixation using tricortical fixation with only one screw [14]. Van den Bekerom et al. managed Weber B fractures with 3.5mm conventional stainless syndesmotic screws inserted 2cm proximally to the syndesmosis and parallel to the tibiotalar joint line, with the ankle in a neutral or slightly dorsiflexed position [16].

## Functional outcomes

Kohake et al. found that patients who required syndesmotic fixation had a lower OMAS than patients who did not (84 vs. 90, p = 0.178) [6]. Between SF and non-SF patients, there were significant differences in FFI, AOFAS, Hannover-Score, SF-36 PCS, SF-36 MCS, VAS scores in the morning (0.3 vs. 0.5, p = 0.463), or VAS scores in the evening (1.2 vs. 1.1, p = 0.843) [6]. Kortekangas et al. found significant improvements in OMAS and VAS (pain) for the non-SF group [5]. Improvements in OMAS, VAS (pain and function), and RAND-36 (physical and pain) had no significant differences between the two groups, and improvements in functional parameters and pain measurements were insignificant within the SF group [5]. Lehtola et al. found that the mean OMAS in SF patients and non-SF patients was 87.3 and 89.0 respectively (p = 0.767) while there were no differences in VAS (pain and function) and RAND 36 [18]. Ntalos et al. found no significant differences in AOFAS and SF-36 scores between SF and non-SF groups [19]. Paez et al. found that the mean FAAM ADL score was 95.61 in the SF group and 98.00 in the non-SF group (p = 0.87) while the mean FAAM Sports score was 91.50 in the SF group and 92.83 in the non-SF group (p = 0.23) [15]. There were also no significant differences in satisfaction scores [15]. Pakarinen et al. found no significant differences in OMAS, VAS (pain and function) or RAND-36 scores [17].

Regarding ROM, Kohake et al. found that, although plantarflexion did not differ significantly between groups, SF patients showed significantly restricted ankle dorsiflexion with a mean difference of 5˚ compared to patients who did not (15˚ vs. 20˚, p = 0.028) [6]. Kortekangas et al. found that ROM improved in both groups but there was no significant difference between the groups [5]. Lehtola et al. found no differences in ROM between SF and non-SF patients [18]. Pakarinen et al. found that between SF and non-SF patients, the ROM after one year was similar with a mean difference of 1˚ in dorsiflexion (p = 0.34) and a mean difference of -6˚ in plantarflexion (p = 0.58) [17]. Sipahioglu et al. found that ROM was sufficient for daily activities in both SF and non-SF groups [14]. A weight-bearing analysis conducted by Kohake et al., identified valgus malalignment of the normal hindfoot axis in 8% of patients, pes planus in 12%, a combination of both in 10%, pes cavus in 8%, splayfoot in 7%, and a hindfoot valgus on the treated side in 50% [6]. There was, however, no significant difference in the

incidence of unilateral changes of the plantar aspect of the weight-bearing foot between the two groups (18% vs. 11%, p = 0.703) [6].

### Radiological outcomes

In two studies, the ankle mortise was congruent in all patients [5, 18]. Ntalos et al. measured tibiofibular alignment on MRI and found no significant differences between SF and non-SF groups [19]. Pakarinen et al. found no differences in mean tibiotalar clear space (3.5mm vs. 3.2mm, p = 0.34), mean tibiofibular clear space (5.4mm vs. 5.5mm, p = 0.41) or talar tilt (one in each group, p>0.09) respectively [17]. Sipahioglu et al. found that the decrease in anteroposterior tibiofibular distance was statistically significant in SF while the difference between pre-operative and post-operative lateral fibular distance and medial mortise distance values was not statistically significant [14].

Osteoarthritis can be defined as the presence of osteophytes with joint space narrowing and/or deformation and was evaluated using the Kellgren-Lawrence Score (KLS) in three studies [5, 6, 18]. Kohake et al. found that 28% of non-SF patients and 38% of SF patients had an increase in KLS of +1 after surgery while both groups had a single patient with an increase of +2, demonstrating no difference in the likelihood of post-traumatic osteoarthrosis [6]. Korte-kangas et al. found that, in the SF group, one patient was graded as KLS class I and 12 patients as KLS class II while in the non-SF group, seven patients were graded KLS class I and two patients graded KLS class II (p = 0.101) [5]. In the study by Lehtola et al., all patients in the SF group were graded KLS class II, one patient having deteriorated from class I to class II between the mid-term follow-up (mean 4.8 years) and final follow-up (mean 9.7 years), while in the non-SF group, one patient was graded KLS class I, five patients as KLS class II, and two as KLS class III [18].

Kortekangas et al. detected syndesmotic calcification in 62% of patients in the SF group compared to 11% of non-SF patients (p = 0.031), joint cartilage defects visible on MRI in 67% of SF patients and 40% of non-SF patients, and no significant differences in TC joint cartilage height anteriorly and posteriorly or posterior facet cartilage height [5].

### Complications

Complications were discussed in four studies [5, 6, 16, 18]. These included broken screws [5, 6, 18], loosened screws left in place [5, 6, 18], postoperative wound infections [6], wound healing problems [6], local irritation requiring screw or plate removal [5, 18], and tibiofibular synostosis [16]. Screw loosening was identified on radiography [5]. In the study by Kohake et al., post-operative wound infections were also seen in 2 non-SF patients (p = 0.602) and wound healing problems in 4 non-SF patients [6]. Paez et al. found that among SF patients in their study, there were two with superficial surgical site infections, one with a stitch abscess and one with postoperative fourth toe numbness while among non-SF patients, there were three with stitch abscesses, one with wound dehiscence requiring irrigation, debridement, and closure, and one patient requiring operative scar revision [15]. However, it is unclear how many patients among these were had Weber B fractures instead of Weber C fractures [15]. Complications among SF patients are summarised in Table 2.

### Limitations of studies selected for review

NOS scores ranged from seven [5, 6, 16] to eight [14, 15, 17–19] (Table 3). Since these studies assessed clinical and radiological outcomes of syndesmotic fixation after several months of follow-up, these outcomes could not have been measured at the start of the studies. As such, regarding selection bias, studies were unable to demonstrate that outcomes of interest were

**Table 2. Complications among SF patients.**

| Author (year) | Screw breakage | Screw loosening (screw left in place) | Infection | Healing problems | Local irritation | Tibiofibular synostosis |
|---|---|---|---|---|---|---|
| Kohake et al., (2019) [6] | 2 | 2 | 2 | 2 | - | - |
| Kortekangas et al., (2014) [5] | 1 | 12 | - | - | 2 (screws removed) 1 (screws and plate removed) | - |
| Lehtola et al., (2022) [18] | 2 | 6 | - | - | 4 (screws removed) | - |
| van den Bekerom et al., (2013) [16] | 0 | 0 | - | - | - | 1 |

absent at the start of each study. Regarding comparability bias and confounding factors, 2 studies controlled for BMI [6, 15], 6 studies controlled for additional fractures and ligamentous injuries [5, 6, 14, 17–19], and 1 study controlled for past operations to the ankle [18]. A loss of <5% to follow-up was deemed acceptable. As such, Kohake et al. posed some risk of outcome bias, having a 37% loss to follow-up [6].

## Discussion

### Syndesmotic stability and fixation

According to the literature, the need to stabilise the syndesmosis is based on the association between instability and early degenerative changes in the tibiotalar joint and poor functional outcomes [18]. Severe ankle fractures with tibiofibular diastasis requiring syndesmotic fixation also risk synostosis as a late complication, usually within three months of the injury [16].

**Table 3. NOS scores.**

| Study | Selection | | | | Comparability | | Outcome | | | Total |
|---|---|---|---|---|---|---|---|---|---|---|
| Cohort studies | Representativeness of the exposed cohort | Selection of the non-exposed cohort | Ascertainment of exposure | Demonstration that outcome of interest was not present at the start of the study | Main factor | Additional factor | Assessment of outcome | Was follow-up long enough for outcomes to occur? | Adequacy of follow-up of cohorts | 9/9 |
| Kohake et al., (2019) [6] | * | * | * | | * | * | * | * | | 7 |
| Kortekangas et al., (2014) [5] | * | * | * | | * | * | * | * | * | 7 |
| Lehtola et al., (2022) [18] | * | * | * | | * | * | * | * | * | 8 |
| Ntalos et al., (2018) [19] | * | * | * | | * | * | * | * | * | 8 |
| Paez et al., (2021) [15] | * | * | * | | * | * | * | * | * | 8 |
| Pakarinen et al., (2011) [17] | * | * | * | | * | * | * | * | * | 8 |
| Sipahioglu et al., (2018) [14] | * | * | * | | * | * | * | * | * | 8 |
| van den Bekerom et al., (2013) [16] | * | * | * | | * | | * | * | * | 7 |

However, negative effects of syndesmosis fixation included a 15–52% risk of malreduction of the distal tibiofibular joint and synostosis or calcification around the tibiofibular joint leading to impaired ankle function [5]. After an ankle fracture is reduced, the two indications for surgical treatment that have been agreed upon are static incongruity and dynamic incongruity or instability [14]. Ankle instability arises when the anterior talofibular and deep deltoid ligaments are torn, leading to a >1mm displacement of the talus on a mortise radiograph compared to the contralateral extremity [14]. The necessity of distal tibiofibular joint transfixion in achieving good outcomes in Weber B fractures has yet to be determined [17]. An evaluation of the accuracy of pre-operative radiographs in predicting syndesmotic injury by Kellam et al. found that Weber B fractures ending between the level of the plafond and the physeal scar were 2.6 times more likely to have a syndesmotic injury than fractures ending below the plafond [20]. In skeletally immature paediatric patients, although syndesmotic injuries are rare because ligaments are stronger than the physeal plate and often cause ankle fractures instead of purely ligamentous injuries, thorough intraoperative testing of the syndesmosis should always be performed especially in rotational ankle fractures [15].

Kohake et al. found no significant difference in functional outcome scores, pain levels, quality of life, or likelihood of post-traumatic osteoarthritis when comparing patients with syndesmotic injury and screw fixation with patients without syndesmotic injury while there was significantly restricted dorsiflexion in ankles with syndesmotic rupture treated with tibiofibular screws, concluding that treating syndesmotic injuries with tibiofibular screws did not have significant effects on the clinical outcomes or quality of life [6]. Kortekangas et al. found that syndesmotic fixation in Weber B fractures did not have significantly different functional or radiological results, risk of osteoarthritis, or development of joint cartilage defects compared to no fixation [5]. Lehtola et al. found that Weber B fractures treated with syndesmotic fixation yielded similar functional, radiological, and quality of life survey results to those without syndesmotic fixation while malleolar fixation without syndesmotic fixation did not lead to widening of the ankle mortise or early degenerative osteoarthritis, concluding that Weber B fractures could be treated with only malleolar fixation and produce excellent outcomes [18]. Ntalos et al. did not identify any correlation between tibiofibular alignment and clinical outcomes at a mid-term follow-up and that tibiotalar congruency and anatomic reduction were more important than tibiofibular reduction where post-traumatic and post-operative were more tolerable in patients [19]. They also observed significant malreduction in bimalleolar/trimalleolar/dislocated type Weber B fractures, demonstrating the significant impact that additional injuries or a fracture-dislocation can have on tibiofibular positioning [19]. Paez et al. found that patients with syndesmotic fixation had equivalent functional outcomes, rates of return to sport and overall satisfaction to those without syndesmotic fixation [15]. They also determined male gender, low BMI, and longer follow-up times to be independent positive predictors of functional outcomes instead of syndesmotic fixation [15]. Pakarinen et al. found that syndesmotic fixation with a screw in Weber B ankle fractures had significant differences in functional outcomes, pain, or radiological parameters compared to no fixation after one year [17]. They also highlight malreduction of the syndesmosis as a negative effect of screw fixation [17]. A study of 56 ankle fractures by Veen and Zuurmond found that patients treated with syndesmotic repair and screw removal at 8 weeks had comparable results to those without syndesmotic injury at the 6 year follow-up [21]. A study of 347 ankle fractures by Egol et al. found that patients treated with syndesmotic stabilization in addition to malleolar fracture fixation had poorer outcomes than patients treated with malleolar fracture fixation alone after one year [22]. However, Sipahioglu et al. concluded that syndesmotic fixation should always be carried out in diastasis, regardless of fracture type [14].

## Alternatives to syndesmotic screws

Alternatives to syndesmotic screws reported in the literature include suture-button devices [23], tightrope fixation, and antiglide fixation [24]. Pavone et al. found that suture-button devices had comparable functional outcomes to screw fixation while having faster recovery times, higher foot dorsiflexion and forefoot supination and adduction [23]. Willmott et al. found that, among four Weber C fractures, one Maisonneuve fracture and one isolated syndesmotic disruption without fracture, tightrope fixation successfully reduced and maintained the distal tibiofibular syndesmosis and allowed the initiation of weightbearing by six weeks [25]. Despite good functional outcomes and patient satisfaction, they suspected that the insertion of a tightrope may stimulate a chronic inflammatory response, resulting in two out of six of these patients needing the tightrope to be removed [25]. A study of 37 ankle fractures including isolated syndesmotic injuries, trimalleolar fractures, bimalleolar fractures, Weber B fractures, Weber C fractures, Salter Harris type 3 fractures, and Maisonneuve fractures by Rigby et al. found that tightrope fixation rarely required removal and enabled physiological motion of the syndesmosis and an early return to weightbearing [26]. DeKeyser et al. propose antiglide fixation as an alternative to distal screw fixation in the management of Weber B fractures with the correct obliquity because antiglide fixation can apply axial forces and harness force vectors of the obliquity to maintain the reduction [24]. Their study comparing antiglide plate fixation with neutralization plate fixation found that patients with neutralization plate fixation and distal screw fixation required hardware to be removed 2.5 times more frequently than patients fixed with antiglide fixation [24]. Rates of soft tissue complications between the two groups were comparable [24].

## Post-traumatic osteoarthritis

The risk of an ankle fusion or arthroplasty due to post-traumatic osteoarthritis is highest within three years of the ankle fracture [18]. Ntalos et al. suggest that the development of osteoarthritis may depend on tibiofibular malreduction in addition to obesity or osteochondral lesions, hence recommending three-dimensional scans to verify syndesmotic malreduction, especially in bimalleolar/trimalleolar/dislocated type Weber B fractures and isolated Weber C fractures with syndesmotic fixation [19]. Van den Bekerom stated that post-traumatic osteoarthritis depended more on fracture reduction, fracture mechanism and initial cartilage lesions than on chronic syndesmotic instability and inadequate distal tibiofibular joint reduction [16].

## Hardware removal

According to the literature, syndesmotic screws should be removed by 8–9 weeks post-operatively [27, 28]. Until screws are removed, some surgeons advise keeping patients non-weightbearing due to rigid syndesmotic fixation disrupting normal fibular motion and mortise widening during ankle movement [16]. However, there is no consensus on the optimal management of syndesmotic screws after initial syndesmotic fixation [4]. According to Tucker et al., hardware is removed in 23% of ankle procedures and there is a 3–20% associated risk of complication with screw removal [4]. Advantages of screw removal include the restoration of physiological biomechanics, the removal of a foreign body and improved long-term outcomes while disadvantages include a risk of recurrent diastasis of up to 11.5% and the need to undergo unnecessary surgery [4]. Kohake et al. found that removing hardware did not produce superior functional, radiological, or mental outcome scores so routine hardware removal does not appear to improve clinical outcomes and leaving hardware in situ could alleviate a burden on hospital resources without any negative effects on patients [6]. Sipahioglu et al. found that syndesmotic screws could be removed as early as 6 weeks post-operatively, observing no

increases in tibiofibular and medial mortise distances at a 3-month follow-up and presuming that 3 months is enough for soft tissue healing [14]. Tucker et al. found no significant difference in functional outcomes between patients with retained screws and those who retained screws (p = 0.135), concluding that screws should remain in situ unless there was a persistent hardware complaint such as irritation whereby, they can be removed 4–6 months post-operatively to avoid the risk of delayed diastasis [4].

## Study limitations

This systematic review used three databases (EMBASE, PubMed, and CINAHL) which may not have included all relevant studies that may have been available on other databases. The search was limited to articles in English and grey literature was not examined so relevant articles that fall under these categories would have been excluded. Some articles that were excluded assessed both Weber B and C fractures but did not specify the outcomes of syndesmotic fixation in Weber B fractures in isolation. As such, these articles that could have contributed relevant data regarding Weber B fractures needed to be left out of this review.

Heterogeneity among studies made performing a meta-analysis unfeasible. Studies included in this review used several different indicators to measure clinical, functional, radiological, and quality of life outcomes in Weber B fractures. As previously mentioned, studies varied in syndesmotic fixation techniques. Evaluation of syndesmotic instability also varied between studies with intraoperative bone hook tests being used in 3 studies [6, 14, 16], external stress tests being used in 5 studies (5,15,17–19), and a direct visualisation of the syndesmosis in 2 studies [14, 19]. The time at which outcome measures were recorded also varied greatly between studies, ranging from 1 year [17] to 9.7 years [18].

## Future research

Our systematic review identified a gap in the literature regarding the outcomes of syndesmotic fixation in Weber B fractures using methods other than syndesmotic screws. Although there are studies that assess the use of suture button devices and tightrope fixation in ankle fractures, outcomes of these studies do not discuss specific outcomes in Weber B fractures. As such, future studies should specifically focus on the outcomes of Weber B fractures to evaluate whether there are any benefits to using these alternative methods of syndesmotic fixation in Weber B fractures.

## Conclusion

Functional, radiological, and quality-of-life outcomes in patients with syndesmotic screws are similar to those of patients not managed with syndesmotic screws. Incidences of post-traumatic osteoarthritis are also similar between the two groups. Only one study was identified to be in favour of syndesmotic fixation in all cases of diastasis. As such, syndesmotic fixation with screws may not be a necessity in the management of Weber B fractures. Screws are also associated with breakage, loosening, local irritation and infections. Suture button devices and anti-glide fixation techniques appear to be valid alternatives to syndesmotic screws. With regards to hardware removal, it is acceptable to leave screws in situ and there is no need for routine hardware removal unless the hardware is causing significant side effects for the patient.

## Supporting information

**S1 Checklist. PRISMA 2020 checklist.**
(DOCX)

## Author Contributions

**Conceptualization:** Brandon Lim, Mohamed Shaalan, Sven O'hEireamhoin, Frank Lyons.

**Data curation:** Brandon Lim, Mohamed Shaalan.

**Formal analysis:** Brandon Lim, Mohamed Shaalan.

**Investigation:** Brandon Lim, Mohamed Shaalan.

**Methodology:** Brandon Lim, Mohamed Shaalan.

**Supervision:** Mohamed Shaalan, Sven O'hEireamhoin, Frank Lyons.

**Writing – original draft:** Brandon Lim.

**Writing – review & editing:** Brandon Lim, Mohamed Shaalan.

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
