## [Decision Letter · Decision Letter 0]

26 Apr 2024

PONE-D-24-02565Syndesmotic fixation in Weber B ankle fractures: a systematic reviewPLOS ONE

Dear Dr. Lim,

Thank you for submitting your manuscript to PLOS ONE. After careful consideration, we feel that it has merit but does not fully meet PLOS ONE’s publication criteria as it currently stands. Therefore, we invite you to submit a revised version of the manuscript that addresses the points raised during the review process.

**ACADEMIC EDITOR: Minor Review**

We look forward to receiving your revised manuscript.

Kind regards,

Raffaele Vitiello

Academic Editor

PLOS ONE

Reviewers' comments:

Reviewer's Responses to Questions

**Comments to the Author**

1. Is the manuscript technically sound, and do the data support the conclusions?

Reviewer #1: Yes

2. Has the statistical analysis been performed appropriately and rigorously? 

Reviewer #1: Yes

3. Have the authors made all data underlying the findings in their manuscript fully available?

Reviewer #1: Yes

4. Is the manuscript presented in an intelligible fashion and written in standard English?

Reviewer #1: Yes

5. Review Comments to the Author

Reviewer #1: I read with particular interest your study aimed at evaluating the current literature on syndesmotic fixation in Weber type B fractures, assessing outcomes and complications of syndesmotic fixation, and evaluating the necessity of syndesmotic fixation in Weber type B fractures. The abstract provides a clear overview of the study's objectives, methods, and main conclusions. The introduction provides a general framework of Weber type B ankle fractures and their management, identifying syndesmotic fixation as a crucial aspect. However, a clear rationale for conducting the systematic review in this specific area is not provided. A more thorough justification would be advantageous to establish the need and relevance of the study. The Materials and Methods section describes in detail the inclusion criteria, search strategy, and approach to assessing study quality. However, a more detailed explanation of the methods used for data analysis from included studies would be beneficial. Additionally, it is not mentioned whether a study selection process was independently conducted by multiple reviewers, which could influence the validity of the results. Discussion of limitations of the included studies is limited. It would be helpful to identify any sources of bias or confounding factors that could affect the results and their interpretation. The discussion section does not exhaustively address the limitations of the study, such as potential heterogeneity among included studies and variability in syndesmotic fixation techniques. Clarifying why this specific systematic review was conducted and why it is relevant to the scientific community would be helpful. A more detailed description of the methods used to analyze data extracted from included studies could improve the transparency and validity of the study. Study limitations, such as potential heterogeneity among included studies and variability in syndesmotic fixation techniques, should be discussed in more detail. Greater attention to these points could improve the completeness and robustness of the study, thereby contributing to better understanding and application of its results.

6. PLOS authors have the option to publish the peer review history of their article (what does this mean?). If published, this will include your full peer review and any attached files.

Reviewer #1: **Yes: **Giuseppe Basile

---

## [Author Response · Author response to Decision Letter 0]

26 Apr 2024

Emily Chenette

Editor-in-Chief

PLOS ONE

April 2024

Manuscript number: PONE-D-24-02565

Manuscript title: Syndesmotic fixation in Weber B ankle fractures: a systematic review

Dear Dr. Chenette and Peer Reviewers:

Thank you for your time and insightful comments. We have attached our point-by-point response to your comments below. Thank you for your consideration.

Comment: I read with particular interest your study aimed at evaluating the current literature on syndesmotic fixation in Weber type B fractures, assessing outcomes and complications of syndesmotic fixation, and evaluating the necessity of syndesmotic fixation in Weber type B fractures. The abstract provides a clear overview of the study's objectives, methods, and main conclusions. The introduction provides a general framework of Weber type B ankle fractures and their management, identifying syndesmotic fixation as a crucial aspect. However, a clear rationale for conducting the systematic review in this specific area is not provided. A more thorough justification would be advantageous to establish the need and relevance of the study.

Comment: Clarifying why this specific systematic review was conducted and why it is relevant to the scientific community would be helpful.

Response: To address the above points regarding the relevance of this review, we have included in the introduction a background regarding the functionality of syndesmotic fixation methods as well as their problems. We also discuss how there appears to be no benefit in terms of clinical outcomes in choosing a surgical management for Weber B fractures compared to non-surgical approaches. There is also a gap in the literature regarding whether there is really a need for syndesmotic fixation in the management of these fractures. Our review thus aims to evaluate studies that have evaluated fixation versus non-fixation in the management of Weber B fractures.

Comment: The Materials and Methods section describes in detail the inclusion criteria, search strategy, and approach to assessing study quality. However, a more detailed explanation of the methods used for data analysis from included studies would be beneficial. Additionally, it is not mentioned whether a study selection process was independently conducted by multiple reviewers, which could influence the validity of the results. A more detailed description of the methods used to analyze data extracted from included studies could improve the transparency and validity of the study.

Response: To address the points regarding data collection and analysis, we have updated the manuscript to clarify that these steps were carried out by two independent reviewers and data collected from studies was compiled independently.

Comment: Discussion of limitations of the included studies is limited. It would be helpful to identify any sources of bias or confounding factors that could affect the results and their interpretation.

Response: To address comments regarding the evaluation of individual studies, we have used the Newcastle Ottowa Scale to evaluate risk of biases. We have clarified in our manuscript the confounding factors that were controlled for in individual studies as well.

Comment: The discussion section does not exhaustively address the limitations of the study, such as potential heterogeneity among included studies and variability in syndesmotic fixation techniques.

Comment: Study limitations, such as potential heterogeneity among included studies and variability in syndesmotic fixation techniques, should be discussed in more detail. Greater attention to these points could improve the completeness and robustness of the study, thereby contributing to better understanding and application of its results.

Response: To address these points, we have further included limitations of our study which was heterogeneity among studies making a meta-analysis unfeasible. Studies used several different indicators to measure clinical, functional, radiological, and quality of life outcomes in Weber B fractures, different syndesmotic fixation techniques, different methods to evaluate syndesmotic instability, and different follow-up times at which outcome measures were recorded.

---

## [Decision Letter · Decision Letter 1]

8 May 2024

Syndesmotic fixation in Weber B ankle fractures: a systematic review

PONE-D-24-02565R1

Dear Dr. Lim,

We’re pleased to inform you that your manuscript has been judged scientifically suitable for publication and will be formally accepted for publication once it meets all outstanding technical requirements.

Kind regards,

Raffaele Vitiello

Academic Editor

PLOS ONE

Additional Editor Comments (optional):

Reviewers' comments:

Reviewer's Responses to Questions

**Comments to the Author**

1. If the authors have adequately addressed your comments raised in a previous round of review and you feel that this manuscript is now acceptable for publication, you may indicate that here to bypass the “Comments to the Author” section, enter your conflict of interest statement in the “Confidential to Editor” section, and submit your "Accept" recommendation.

Reviewer #1: All comments have been addressed

2. Is the manuscript technically sound, and do the data support the conclusions?

Reviewer #1: Yes

3. Has the statistical analysis been performed appropriately and rigorously? 

Reviewer #1: Yes

4. Have the authors made all data underlying the findings in their manuscript fully available?

Reviewer #1: Yes

5. Is the manuscript presented in an intelligible fashion and written in standard English?

Reviewer #1: Yes

6. Review Comments to the Author

Reviewer #1: Dear Authors, I have read your additions, finding them entirely consistent and in accordance with my expectations. I consider your work interesting. The purpose is clear and respected. I believe that the information provided is to be considered entirely sufficient and represents useful elements to encourage the development of new scientific work.

7. PLOS authors have the option to publish the peer review history of their article (what does this mean?). If published, this will include your full peer review and any attached files.

Reviewer #1: **Yes: **Giuseppe Basile

---

## [Editor Report · Acceptance letter]

13 May 2024

PONE-D-24-02565R1 

PLOS ONE

Dear Dr. Lim, 

I'm pleased to inform you that your manuscript has been deemed suitable for publication in PLOS ONE. Congratulations! Your manuscript is now being handed over to our production team.

Kind regards, 

on behalf of

Dr. Raffaele Vitiello 

Academic Editor

PLOS ONE